# AFFIN-SPACE: Learning Affine-Invariant Representations for 3D Spatial Understanding with MLLMs

Zhenyu Lu [1 2]  Liupeng Li [3 1]  Jinpeng Wang [3]  Haoqian Kang [3]  Manyuan Zhang [4]  Yan Feng [4]
Ke Chen [1]  Yaowei Wang [3 1]

## Abstract

While MLLMs show promising capacity on general visual understanding, they suffer from *geometric fragility*: standard visual representations often degrade rapidly under changes in viewpoint and viewing distance. Our analysis identifies that existing paradigms, whether relying on input-level fusion or latent reconstruction, remain entangled with the view-dependent pixel grid, failing to decouple intrinsic 3D structure from extrinsic camera pose. To address this, we introduce **AFFIN-SPACE**, a framework that enforces strict affine invariance to enable robust spatial understanding. Unlike implicit learning approaches, AFFIN-SPACE introduces a two-stage explicit decoupling mechanism. First, it employs *explicit geometric resampling* by utilizing decomposed affine quantities (derived from pose features) to spatially align 3D features to a canonical state before fusion. Second, within the MLLM, we implement *affine-invariant constraints* via an orthogonal projection mechanism, which mathematically strips away pose-dependent noise from the hidden states while retaining recoverable geometric semantics through conditional reconstruction. Extensive experiments on VSI-Bench, ScanQA, SQA3D, Scan2Cap, and EmbodiedScan demonstrate that AFFIN-SPACE achieves state-of-the-art performance. Crucially, our approach exhibits superior stability against affine perturbations, validating the effectiveness of explicitly modeling geometric invariance for complex spatial tasks. Code is released at https://github.com/ZhenyuLU-Heliodore/AffIn-Space.

[1]Peng Cheng Laboratory [2]Shenzhen Institutes of Advanced Technology, Chinese Academy of Sciences [3]Harbin Institute of Technology, Shenzhen [4]Meituan. Correspondence to: Jinpeng Wang <wangjp26@gmail.com>, Yaowei Wang <wangyaowei@hit.edu.cn>.

*Proceedings of the 43 $^{rd}$ International Conference on Machine Learning*, Seoul, South Korea. PMLR 306, 2026. Copyright 2026 by the author(s).

## 1. Introduction

Reliable 3D spatial understanding demands geometric stability. While a scene's intrinsic 3D structure remains constant, its 2D projection varies dramatically with viewpoint, manifesting as complex affine transformations. Achieving robust spatial intelligence thus requires affine invariance, which defines the capacity to decouple consistent geometry from varying appearances. However, current Multimodal Large Language Models (MLLMs) (Chen et al., 2024c; Bai et al., 2025; Li et al., 2025a) fundamentally lack this property, resulting in representations that are *viewpoint-dependent* and inconsistent across views.

Existing approaches to equip MLLMs with geometric priors largely fall into two categories, yet both struggle to address this fundamental challenge. The first category employs input-level fusion, mixing geometry-aware features (Wang et al., 2025a) with visual tokens. While architectural changes are minimal, it offers no mechanism to decouple intrinsic geometry from extrinsic pose. The second category introduces learned latent tokens trained with a reconstruction objective. However, we identify an inherent limitation in these paradigms, namely the geometric fragility. Without explicit invariance constraints, these models tend to overfit to continuous visual patterns in dense video streams rather than establishing a robust structural understanding (Liu et al., 2022). Consequently, their learned representations remain entangled with the specific input pixel grid, failing to generalize when the viewpoint (Shi et al., 2024) undergoes large geometric transformations.

This fragility is empirically evident in our preliminary analysis. As illustrated in Figure 2a, methods relying on simple feature fusion (e.g., LLaVA-Video (Zhang et al., 2025d)) struggle to sustain spatial understanding when view density decreases, exhibiting a significant performance gap (up to $1.4\times$) compared to our geometrically grounded model. Furthermore, the representations degrade under affine perturbations (Morel & Yu, 2009). As shown in Figure 2b, even approaches fused with geometric representations suffer from a sharp performance drop (widening to a $1.5\times$ gap) as distortion intensity increases. This decline confirms that standard representations are constrained to the view-

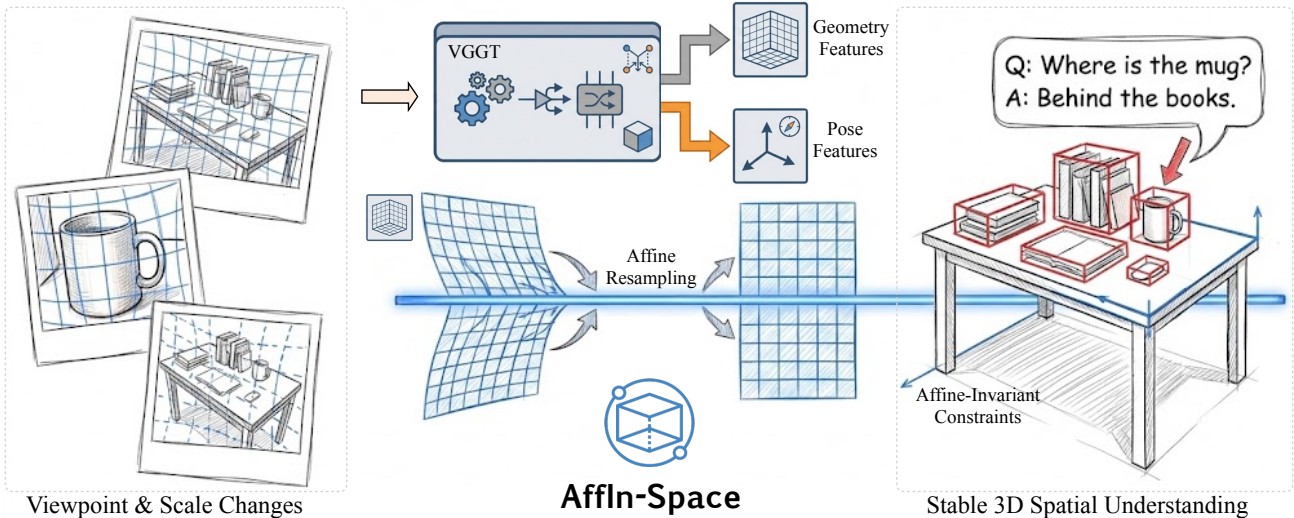

**Figure 1.** **Enhancing 3D MLLMs with Affine Invariance.** While standard visual models often struggle with inconsistent spatial relationships due to viewpoint shifts (left), AFFIN-SPACE utilizes 3D foundation model features (VGGT) and explicit geometric resampling to build a stable latent space. This ensures the model's understanding (right) remains consistent and accurate across different scales and angles, facilitating tasks like 3D spatial QA and object detection.

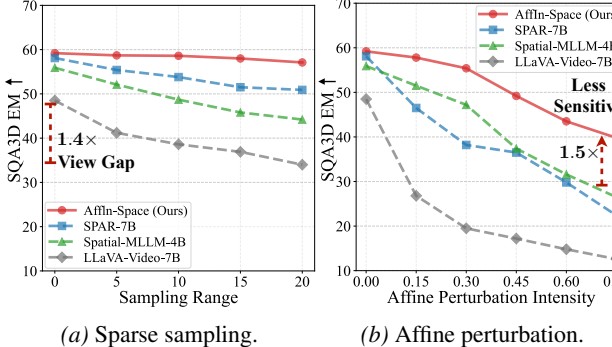

*(a) Sparse sampling.*     *(b) Affine perturbation.*

**Figure 2.** **Stability and geometric sensitivity.** We compare SPAR-7B, Spatial-MLLM-4B, and LLaVA-Video-7B with our AFFIN-SPACE. (a) Performance across sampling ranges shows AFFIN-SPACE maintains a consistent $1.4\times$ View Gap over the video baseline. (b) AFFIN-SPACE exhibits minimal sensitivity to affine distortions, demonstrating a gradual decay compared to the $1.5\times$ performance drop seen in models lacking geometric priors.

dependent (Li et al., 2025b) pixel domain, necessitating a shift toward explicit affine modeling.

To address this, we introduce **AFFIN-SPACE**, a framework that enforces strict **Aff**ine **In**variance to support robust understanding in the 3D **Space**. Unlike implicit approaches, we propose a mechanism to mathematically decouple geometry from pose via two key stages. First, we introduce **explicit geometric resampling**: AFFIN-SPACE utilizes pose features to predict decomposed affine quantities (scale, rotations, tilt) (Morel & Yu, 2009) and explicitly resamples the 3D features to a canonical state before fusion. This ensures that geometric variations are handled explicitly at

the input level. Second, within the MLLM, we enforce **affine-invariant constraints** via an orthogonal projection mechanism. By projecting hidden states onto the orthogonal complement (Chang et al., 2025) of the subspace spanned by the affine parameters, we mathematically remove pose-dependent noise from the semantic representation. Finally, to prevent feature collapse, the model reconstructs pooled 3D features conditioned on these invariant states, ensuring the representation retains recoverable geometric semantics.

Across standard benchmarks, AFFIN-SPACE achieves results at or above the previous best on 3D spatial question answering (QA) benchmarks, including VSI-Bench (Yang et al., 2025), ScanQA (Azuma et al., 2022) and SQA3D (Ma et al., 2022), and consistently improves 3D captioning and 3D detection on Scan2Cap (Chen et al., 2021) and EmbodiedScan (Wang et al., 2024). Beyond overall scores, our qualitative and quantitative analyses consistently indicate that learning affine-invariant representations significantly improves cross-view consistency. We further conduct extensive ablations to quantify the contribution of our explicit resampling and invariant constraints, validating their role in stabilizing spatial understanding.

Our contributions are summarized as follows:

- ***Affine-Invariant Learning Framework for 3D MLLMs.*** We propose AFFIN-SPACE, a framework designed to learn affine-invariant representations. By shifting from implicit learning to explicit modeling, it significantly enhances robustness in 3D spatial understanding.
- ***Explicit Geometric Resampling.*** We introduce a mechanism that utilizes pose features to predict decomposed

affine quantities and applies spatial resampling to 3D features, allowing the model to explicitly counteract geometric distortions before multimodal fusion.

- *Affine-Invariant Constraints.* We enforce strict invariance via an orthogonal projection mechanism and couple it with a conditional reconstruction objective. This provides a principled method to decouple pose noise while retaining recoverable geometric semantics.
- *Performance and Robustness.* AFFIN-SPACE achieves state-of-the-art performance on 3D spatial QA, 3D captioning, and 3D detection benchmarks, showing minimal sensitivity to viewpoint changes and affine perturbations compared to prior arts.

## 2. Related Works

**3D Spatial Understanding.** Recent advances in Multimodal Large Language Models (MLLMs) have significantly expanded their capabilities from 2D image perception (Lu et al., 2026a;b) to complex 3D spatial understanding. Existing approaches can be categorized by their reliance on geometric priors. For Explicit 3D-Input Models, early works inject explicit 3D data into LLMs. 3D-LLM (Hong et al., 2023) injects 3D features from point clouds into LLMs by training on rendered point cloud data, while methods like Chat-Scene (Huang et al., 2024), LL3DA (Chen et al., 2024a), and 3D-LLaVA (Deng et al., 2025b) leverage pretrained 3D encoders to handle point clouds or depth maps directly. While effective, their strict reliance on high-quality 3D sensors severely limits applicability in general scenarios where only RGB videos are available. For Video-Based Spatial Models, recent research focuses on learning spatial representations directly from multi-view images to bridge the modality gap. Generalist models like BAGEL (Deng et al., 2025a) unify modalities through massive pre-training. Specialized approaches introduce geometry priors via intermediate supervision. Video-3D LLM (Zheng et al., 2025c) learns position-aware features, and VG-LLM (Zheng et al., 2025b) fuses estimated depth cues. More recently, SPAR (Zhang et al., 2025a) and 3DThinker (Chen et al., 2025b) employ latent reconstruction objectives to implicitly align video features with 3D foundation models. Similarly, GS-Reasoner (Chen et al., 2026) proposes a dualpath pooling mechanism to align geometric features for robust grounding. Despite these advancements, existing video-based methods typically handle geometric variations implicitly, either through early fusion of cues or latent reconstruction losses. As shown in our analysis, this implicit alignment degrades rapidly under viewpoint changes. In contrast, AFFIN-SPACE introduces an explicit affine modeling mechanism. By predicting decomposed affine parameters and enforcing affine-invariant constraints, our framework ensures geometric stability across diverse viewpoints, enabling robust 3D spatial understanding without ground-truth 3D inputs.

**Visual Geometry Models.** The field of 3D geometry estimation is shifting from optimization-based pipelines to data-driven large reconstruction models. FLARE (Zhang et al., 2025c) pioneers this by estimating camera poses and geometry from uncalibrated sparse views in a feed-forward manner. VGGT (Wang et al., 2025a) further unifies this paradigm, employing a transformer architecture to jointly infer camera parameters, point tracks, and dense depth from video sequences. Subsequent works address dynamic and open-world complexities. ViPE (Huang et al., 2025) enables pose estimation for diverse in-the-wild camera models, while $\pi^3$ (Wang et al., 2025b) introduces permutation equivariance to eliminate the dependency on fixed reference frames. In this work, we leverage VGGT not merely for reconstruction, but as a robust 3D encoder. Crucially, beyond using its geometric features as static inputs. We exploit the predicted pose features to model the *transformation* between views, addressing the viewpoint sensitivity that limits current spatial MLLMs.

## 3. Method

We first present the overall architecture and data flow of AFFIN-SPACE in Section 3.1, then detail affine transform learning and the affine-invariant constraints in Section 3.2, and finally describe the learning objectives in Section 3.3.

### 3.1. Overview of AffIn-Space

**Overall.** As shown in Figure 3, given an input sequence of $S$ frames $\boldsymbol{F}$, AFFIN-SPACE encodes $\boldsymbol{F}$ with a dual-path feature encoder to obtain visual representations $\boldsymbol{r}_{\text{vis}}$, 3D geometry representations $\boldsymbol{r}_{\text{geo}}$, and per-frame pose features $\{\boldsymbol{p}_i\}_{i=1}^{S}$. Conditioned on $\boldsymbol{p}_i$, an affine matrix estimator predicts affine parameters $(\boldsymbol{A}_i, \boldsymbol{b}_i)$, and resamples $\boldsymbol{r}_{\text{geo}}$ along spatial dimensions to produce $\tilde{\boldsymbol{r}}_{\text{geo}}$. We assign one special `<INV>` token to each frame, whose hidden states are denoted by $\{\boldsymbol{h}_i\}_{i=1}^{S}$. After the connector, $\tilde{\boldsymbol{r}}_{\text{geo}}$ and $\boldsymbol{r}_{\text{vis}}$ are injected into the MLLM together with the `<INV>` tokens and the question tokens. The MLLM decodes the final outputs, while the `<INV>` tokens are optimized under affine-invariant constraints to obtain affine-invariant representations $\boldsymbol{r}_{\text{inv}}$, which are further used to reconstruct $\tilde{\boldsymbol{r}}_{\text{geo}}$.

**Dual-Path Feature Encoding.** AFFIN-SPACE uses a dual-path feature encoder consisting of a 2D visual encoder $\mathcal{E}_{2\text{d}}$ and a 3D geometric encoder $\mathcal{E}_{3\text{d}}$. We use Qwen2.5-VL (3B/7B) as the MLLM backbone, and use its visual encoder as $\mathcal{E}_{2\text{d}}$. The 2D branch encodes $\boldsymbol{F}$ into visual representations $\boldsymbol{r}_{\text{vis}} = \mathcal{E}_{2\text{d}}(\boldsymbol{F})$. In parallel, we use VGGT as the 3D geometric encoder $\mathcal{E}_{3\text{d}}$, producing $\boldsymbol{r}_{\text{geo}}$ and per-frame pose features $\{\boldsymbol{p}_i\}_{i=1}^{S}$, i.e.,

$$\left(\boldsymbol{r}_{\text{geo}}, \{\boldsymbol{p}_i\}_{i=1}^{S}\right) = \mathcal{E}_{3\text{d}}(\boldsymbol{F}). \tag{1}$$

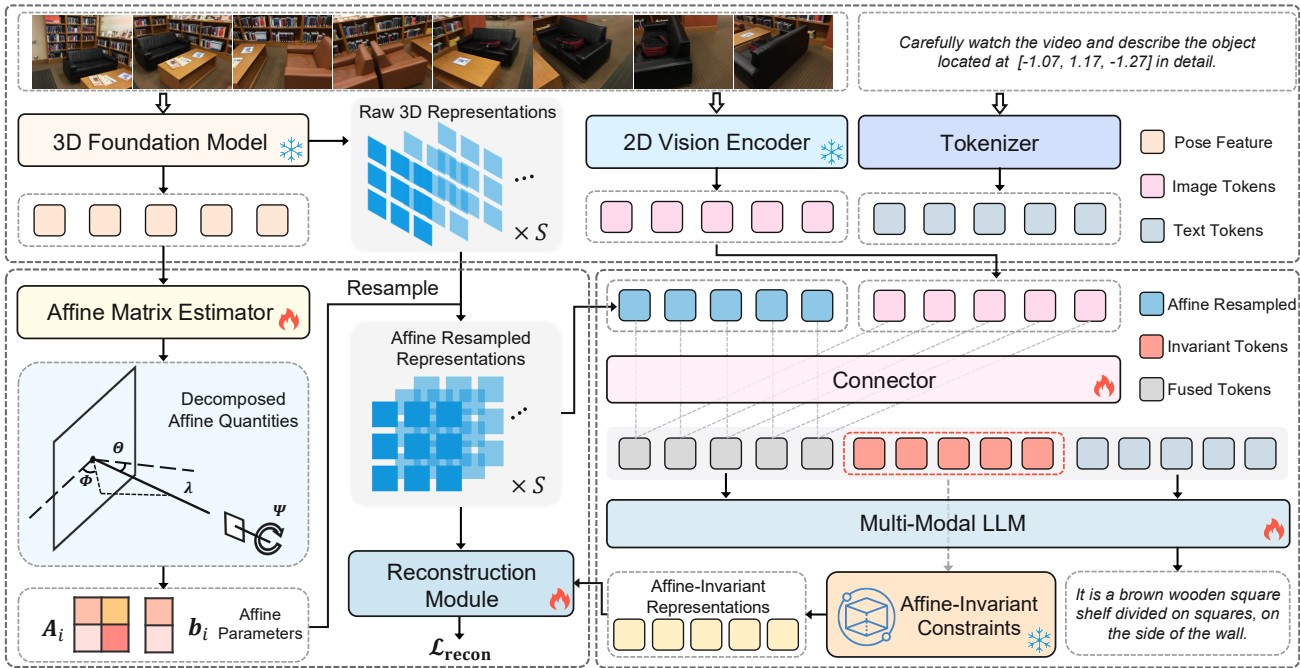

*Figure 3.* **Overview of AFFIN-SPACE.** Given video frames, we employ a dual-path pipeline: a frozen 3D Foundation Model (VGGT) extracts geometric $r_{\text{geo}}$ and pose features $p_i$, while a 2D Vision Encoder extracts visual tokens. **Lower Left:** The Affine Matrix Estimator predicts decomposed affine quantities (rotation, tilt, scale) from $p_i$ to compose affine parameters $(A_i, b_i)$, which are used to spatially resample $r_{\text{geo}}$ into an aligned state $\tilde{r}_{\text{geo}}$. **Lower Right:** The resampled features are fused with visual and text tokens via a Connector. Within the MLLM, we learn special `<INV>` tokens subject to strict Affine-Invariant Constraints (via orthogonal projection) and a Reconstruction Module that recovers the pooled 3D features.

**Affine Transform and Resampling.** To ensure the predicted affine transform is physically meaningful and lies in a valid range, we predict four scalars with clear physical interpretations: scale, rotation, tilt, and rotation. These quantities are then composed into the affine parameters $\{(A_i, b_i)\}_{i=1}^{S}$ (specified in Equation (4)). With $(A_i, b_i)$, we apply a differentiable affine resampling to the spatial dimensions of the 3D geometry representations. Concretely, for the $i$-th frame, we resample $r_{\text{geo}}^{(i)}$ on an affinely transformed grid parameterized by $(A_i, b_i)$. Since the transformed coordinates may be non-integer, we use an interpolation kernel $\kappa(\cdot)$ to obtain the resampled representation $\tilde{r}_{\text{geo}}^{(i)}$:

$$\tilde{r}_{\text{geo}}^{(i)}(x) = \sum_{u \in \Omega} r_{\text{geo}}^{(i)}(u)\, \kappa\big(u - (A_i x + b_i)\big), \quad (2)$$

where $x$ indexes the target spatial grid and $\Omega$ denotes the discrete spatial lattice of $r_{\text{geo}}^{(i)}$. In practice, $\kappa$ is instantiated as a standard interpolation kernel.

**Decoding and Reconstruction.** Given the fused multimodal inputs, the MLLM performs autoregressive decoding conditioned on the question and the encoded features from $F$, where we denote the per-frame `<INV>` token hidden states as $h = \{h_i\}_{i=1}^{S}$ and include $h$ in the conditioning

context. Formally, the decoded output sequence $o$ follows

$$P(o \mid Q, F, h) = \prod_t P\Big(o^{(t)} \mid o^{(1:t-1)}, Q, F, h\Big), \quad (3)$$

where $o^{(t)}$ denotes the $t$-th output token. In addition to driving decoding, each `<INV>` representation $h_i$ is constrained, conditioned on the per-frame affine parameters $(A_i, b_i)$, to obtain an affine-invariant representation $r_{\text{inv}}^{(i)}$ that satisfies the strict affine-invariant constraints specified in Equation (7). The resulting $r_{\text{inv}}$ is then fed into a reconstruction head to reconstruct the pooled representation of $\tilde{r}_{\text{geo}}$ to prevent representation collapse and preserve recoverable geometric semantics, where the loss is in Equation (10).

### 3.2. Learning Affine-Invariant Representations

**Predicted Quantity Composition.** Following ASIFT (Morel & Yu, 2009), we decompose the linear part of the affine transform into four quantities (two rotations, one tilt, and one scale) with explicit range constraints, and predict the translation $b_i$ separately. Given the pose feature $p_i$, we predict the two rotation angles via their $(\cos, \sin)$ vectors, as well as tilt, scale, and translation $b_i$, each with a dedicated MLP head. The predicted rotation vectors are $\ell_2$-normalized to form valid $(\cos, \sin)$ pairs. For tilt and scale, we enforce bounded ranges through a

log-space $\mathrm{tanh}$ parameterization: the tilt is constrained to $[1/t_{\max}, t_{\max}]$ with $t_{\max} = 5.8$ (ASIFT default), and the scale is constrained to $[1/\lambda_{\max}, \lambda_{\max}]$ with $\lambda_{\max} = 4.0$. We initialize the head to start from the identity transform by setting the rotation vectors to $(1, 0)$ and the remaining quantities to yield unit tilt/scale and zero translation.

Finally, we compose the affine parameters $(\boldsymbol{A}_i, \boldsymbol{b}_i)$ from the predicted quantities as

$$\boldsymbol{A}_i = \lambda_i \boldsymbol{R}(\psi_i) \boldsymbol{T}(t_i) \boldsymbol{R}(\phi_i), \quad (4)$$

where $\lambda_i$ is the scale, $t_i$ is the tilt, $\phi_i, \psi_i$ are the two rotations, and

$$\boldsymbol{T}(t_i) = \begin{bmatrix} t_i & 0 \\ 0 & 1 \end{bmatrix}, \quad \boldsymbol{R}(\theta) = \begin{bmatrix} \cos\theta & -\sin\theta \\ \sin\theta & \cos\theta \end{bmatrix}. \quad (5)$$

The translation is given by $\boldsymbol{b}_i$.

**Affine-Invariant Constraints.** We enforce affine invariance on the `<INV>` representation $\boldsymbol{h}_i$ by removing a subspace induced by the affine parameters $(\boldsymbol{A}_i, \boldsymbol{b}_i)$. Concretely, we concatenate $(\boldsymbol{A}_i, \boldsymbol{b}_i)$ as the conditioning input to a lightweight MLP to produce an initial rank-$r$ basis $\tilde{\boldsymbol{B}}^{(i)} \in \mathbb{R}^{D \times r}$, where $r \ll D$ (practically $r = 8$) and $D$ is the dimension of $\boldsymbol{r}_{\mathrm{inv}}^{(i)}$. We then apply column-wise Gram–Schmidt orthogonalization to $\tilde{\boldsymbol{B}}^{(i)}$ to obtain an orthonormal basis $\boldsymbol{B}^{(i)}$ satisfying $\left(\boldsymbol{B}^{(i)}\right)^\top \boldsymbol{B}^{(i)} = \boldsymbol{I}$, i.e., the columns of $\boldsymbol{B}^{(i)}$ are orthonormal and span an $r$-dimensional subspace. Finally, we define the affine-invariant representation by the orthogonal-complement projection

$$\boldsymbol{r}_{\mathrm{inv}}^{(i)} = \left(\boldsymbol{I} - \boldsymbol{B}^{(i)}\left(\boldsymbol{B}^{(i)}\right)^\top\right) \boldsymbol{h}_i. \quad (6)$$

By construction,

$$\left(\boldsymbol{B}^{(i)}\right)^\top \boldsymbol{r}_{\mathrm{inv}}^{(i)} = \boldsymbol{0}, \quad (7)$$

which instantiates our strict affine-invariant constraint.

### 3.3. Learning Objectives

We train AFFIN-SPACE with a joint objective that combines autoregressive language modeling and geometric reconstruction. The reconstruction term is introduced to encourage the affine-invariant representations $\boldsymbol{r}_{\mathrm{inv}}^{(i)}$ to remain semantically informative under the strict affine-invariant constraints, i.e., they should retain sufficient information to recover a compact geometric representation rather than collapsing to a uninformative embedding. Overall, we minimize

$$\mathcal{L} = \mathcal{L}_{\mathrm{CE}} + \lambda \mathcal{L}_{\mathrm{RECON}}. \quad (8)$$

The autoregressive decoding objective follows the standard cross-entropy form:

$$\mathcal{L}_{\mathrm{CE}} = -\sum_t \log P\left(o^{(t)} \mid \boldsymbol{o}^{(1:t-1)}, Q, \boldsymbol{F}, \boldsymbol{h}\right). \quad (9)$$

For geometric reconstruction, we supervise $\boldsymbol{r}_{\mathrm{inv}}^{(i)}$ to reconstruct the mean-pooled representation of $\tilde{\boldsymbol{r}}_{\mathrm{geo}}^{(i)}$. We express mean pooling as an expectation over the discrete spatial lattice $\Omega$, and use a lightweight MLP head $\phi(\cdot)$ to map $\boldsymbol{r}_{\mathrm{inv}}^{(i)}$ into the geometric feature space:

$$\mathcal{L}_{\mathrm{RECON}} = \frac{1}{S} \sum_{i=1}^{S} \left\| \phi\left(\boldsymbol{r}_{\mathrm{inv}}^{(i)}\right) - \mathbb{E}_{\boldsymbol{u} \in \Omega}\left[\tilde{\boldsymbol{r}}_{\mathrm{geo}}^{(i)}(\boldsymbol{u})\right] \right\|_2^2. \quad (10)$$

## 4. Experiments

### 4.1. Experimental Setup

**Backbones and Training Data.** We instantiate AFFIN-SPACE with Qwen2.5-VL-3B and Qwen2.5-VL-7B (Bai et al., 2025) as the MLLM backbones, and use VGGT-1B (Wang et al., 2025a) as the 3D geometric encoder. Following VG-LLM, we train AFFIN-SPACE on a mixed dataset that incorporates SPAR-7M (Zhang et al., 2025a) to better leverage the 3D knowledge inherent in the 3D geometric encoder, and the LLaVA-Hound (Zhang et al., 2025b) split of LLaVA-Video-178K (Zhang et al., 2025d) to maintain general video understanding capability. Specifically, we sample 234K instances from SPAR-7M and 63K instances from LLaVA-Hound (3% and 25% of the original datasets, respectively), and further include the training sets of EmbodiedScan (Wang et al., 2024), Scan2Cap(Chen et al., 2021), and SQA3D (Ma et al., 2022).

**Metrics.** For VSI-Bench (Yang et al., 2025), we report accuracy for multiple-choice tasks and Mean Relative Accuracy (MCA) for numerical tasks. MCA calculates the average accuracy across a range of confidence thresholds, where a prediction is considered correct if its relative error is within a specified threshold. For ScanQA and Scan2Cap, we report CIDEr, BLEU-4 (B-4), METEOR, and ROUGE, and we additionally report exact-match accuracy (EM) for both ScanQA and SQA3D. For EmbodiedScan, we follow the monocular detection setting and report the average F1 score per category.

**Implementation Details.** For each experiment, we train on 8 NVIDIA A100 (80GB) GPUs with a batch size of 192. The default hidden layer index for representation alignment is 8. We use AdamW (Loshchilov & Hutter, 2019) with $\beta_1 = 0.9, \beta_2 = 0.999$, weight decay $1 \times 10^{-2}$, and gradient clipping with max norm 1.0. The base learning rate for the MLLM backbone is $8 \times 10^{-6}$, and we apply $15\times$ learning rates to other newly initialized modules. We adopt a constant learning-rate schedule with a warmup ratio of 0.03. The reconstruction loss coefficient is set to 1.0. Before feeding images into the MLLM, we resize frames to match the spatial size of the first frame. We enforce a minimum of 3,136 pixels per image and cap the maximum pixels so that each image corresponds to at most 196 visual tokens, i.e.,

*Table 1.* **Comparison on VSI-Bench.** AFFIN-SPACE outperforms both proprietary APIs and specialized spatial models.

| Methods | Year | Rank | Average | Numerical Question | | | | Multiple-Choice Question | | | |
| --- | --- | --- | --- | --- | --- | --- | --- | --- | --- | --- | --- |
| | | | | Obj. Cnt. | Abs. Dist. | Obj. Size | Room Size | Rel. Dist. | Rel. Dir. | Route Plan | Appr. Order |
| *Proprietary Models (API)* | | | | | | | | | | | |
| GPT-4o | 2024 | – | 34.0 | 46.2 | 5.3 | 43.8 | 38.2 | 37.0 | 41.3 | 31.5 | 28.5 |
| Gemini-1.5-Flash | 2024 | – | 42.1 | 49.8 | 30.8 | 53.5 | 54.4 | 37.7 | 41.0 | 31.5 | 37.8 |
| Gemini-1.5 Pro | 2024 | – | 45.4 | 56.2 | 30.9 | 64.1 | 43.6 | 51.3 | 46.3 | 36.0 | 34.6 |
| Gemini-2.0 Flash | 2025 | – | 45.4 | – | – | – | – | – | – | – | – |
| *Open-Source Models* | | | | | | | | | | | |
| InternVL2-2B | 2024c | 6 | 27.4 | 21.8 | 24.9 | 22.0 | 35.0 | 33.8 | 44.2 | 30.5 | 7.1 |
| InternVL2-40B | 2024c | 5 | 36.0 | 34.9 | 26.9 | 46.5 | 31.8 | 42.1 | 32.2 | 34.0 | 39.6 |
| LongVILA-8B | 2025a | 7 | 21.6 | 29.1 | 9.1 | 16.7 | 0.0 | 29.6 | 30.7 | 32.5 | 25.5 |
| LLaVA-OneVision-72B | 2025a | 3 | 40.2 | 43.5 | 23.9 | 57.6 | 37.5 | 42.5 | 39.9 | 32.5 | 44.6 |
| LLaVA-Video-72B | 2025d | 2 | 40.9 | 48.9 | 22.8 | 57.4 | 35.3 | 42.4 | 36.7 | 35.0 | 48.6 |
| Qwen2.5VL-72B | 2025 | 4 | 37.0 | 25.1 | 29.3 | 54.5 | 38.8 | 38.2 | 37.0 | 34.0 | 28.9 |
| BAGEL-7B-FT | 2025a | 1 | 46.3 | 62.8 | 36.3 | 56.4 | 49.7 | 46.1 | 49.4 | 26.8 | 43.1 |
| *Specialized Spatial Models* | | | | | | | | | | | |
| SAT-LLaVA-Video-7B | 2025a | – | – | – | – | – | 47.3 | 41.1 | 37.1 | *36.1* | 40.4 |
| SPAR-8B | 2025a | 8 | 41.1 | – | – | – | – | – | – | – | – |
| VG-LLM-4B | 2025b | 7 | 47.3 | 66.0 | *37.8* | 55.2 | *59.2* | 44.6 | 45.6 | 33.5 | 36.4 |
| VG-LLM-8B | 2025b | 5 | 50.7 | *67.9* | 37.7 | 58.6 | 62.0 | *46.6* | 40.7 | 32.4 | **59.2** |
| 3DThinker-SFT-4B | 2025b | 4 | 53.2 | – | – | – | – | – | – | – | – |
| 3DThinker-SFT-8B | 2025b | 2 | 57.3 | – | – | – | – | – | – | – | – |
| Spatial-MLLM-4B | 2025 | 6 | 48.4 | 65.3 | 34.8 | 63.1 | 45.1 | 41.3 | **46.2** | 33.5 | 46.3 |
| AFFIN-SPACE-4B | 2026 | 3 | *53.8* | 71.0 | 46.5 | *62.8* | *61.3* | 50.4 | 44.2 | *39.3* | *54.5* |
| AFFIN-SPACE-8B | 2026 | 1 | **57.7** | **75.1** | **49.6** | **68.7** | **67.2** | **53.1** | *45.5* | **43.7** | 59.0 |

the maximum pixel budget is $196 \times 28 \times 28$. We set the maximum sequence length to 2,560.

## 4.2. 3D Spatial QA Results

**Baselines.** To comprehensively assess the spatial understanding capabilities of AFFIN-SPACE, we compare it against a wide range of state-of-the-art methods categorized into three groups. For Proprietary MLLMs, we include leading commercial APIs including GPT-4o (Hurst et al., 2024), and the Gemini-1.5/2.0 family (Team et al., 2024), which serve as strong general-purpose baselines. For Open-Source Video MLLMs, we consider general video-understanding models such as Qwen2.5-VL (Bai et al., 2025), LLaVA-OneVision (Li et al., 2025a), and LLaVA-Video (Zhang et al., 2025d), as well as spatial-specialized video MLLMs like Spatial-MLLM (Wu et al., 2025), BAGEL (Deng et al., 2025a), and SPAR (Zhang et al., 2025a). For 3D/2.5D-Input Models, we compare with methods that explicitly require auxiliary geometric inputs during inference, such as 3D-LLM (Hong et al., 2023), Chat-Scene (Huang et al., 2024), 3D-LLaVA (Deng et al., 2025b), and Video-3D LLM (Zheng et al., 2025c).

**Results.** We evaluate AFFIN-SPACE on three demanding benchmarks: VSI-Bench, SQA3D, and ScanQA. On **VSI-Bench**, as shown in Table 1, AFFIN-SPACE-8B achieves the state-of-the-art performance with an average score of 57.7. Notably, it outperforms the previous best specialized model,

3DThinker-SFT-8B, and significantly surpasses proprietary giants like Gemini-1.5 Pro and GPT-4o. This empirically validates that our explicit affine modeling effectively preserves metric information and spatial consistency, which are often lost in standard video MLLMs. On **ScanQA** and **SQA3D**, as illustrated in Table 2, AFFIN-SPACE demonstrates superior performance among all video-input models. On the generative ScanQA benchmark, our model sets a new record for video-based methods with a CIDEr score of 99.4, surpassing the previous best Spatial-MLLM by a large margin. On SQA3D, AFFIN-SPACE-8B achieves an Exact Match (EM) score of 59.2, outperforming strong competitors like SPAR-7B and BAGEL-7B-FT. Crucially, AFFIN-SPACE competes favorably even against models that utilize heavy 3D priors. For instance, on SQA3D, our method outperforms Video-3D LLM and Chat-Scene, despite AFFIN-SPACE not requiring any depth maps or point clouds. These results suggest that learning affine-invariant representations is a highly effective strategy for 3D spatial understanding, bridging the gap between pure vision-based models and 3D-dependent systems.

## 4.3. 3D Captioning and Detection Results

We further evaluate the capability of AFFIN-SPACE to generate grounded descriptions and perform precise object localization in 3D scenes using Scan2Cap and EmbodiedScan.

**Baselines.** For **Scan2Cap**, we compare AFFIN-SPACE

*Table 2.* **Comparison on ScanQA and SQA3D.** AFFIN-SPACE demonstrates robust performance, achieving results comparable to or better than prior baselines across multiple metrics.

| Methods | ScanQA | | | | | SQA3D |
|---|---|---|---|---|---|---|
| | B-4 | Rouge | CIDEr | Meteor | EM | EM |
| *3D/2.5D-Input Models* | | | | | | |
| 3D-LLM (2023) | 12.0 | 35.7 | 69.4 | 14.5 | 20.5 | – |
| LL3DA (2024a) | 13.5 | 37.3 | 76.8 | 15.9 | – | – |
| Chat-Scene (2024) | 14.3 | 41.6 | 87.7 | 18.0 | 21.6 | 54.6 |
| 3D-LLaVA (2025b) | 17.1 | 43.1 | 92.6 | 18.4 | – | 54.5 |
| Video-3D LLM (2025c) | 16.2 | 49.0 | 102.1 | 19.8 | – | 58.6 |
| *Video-Input Models* | | | | | | |
| Qwen2.5-VL-3B (2025) | 3.8 | 25.4 | 47.4 | 9.7 | – | 43.4 |
| Qwen2.5-VL-7B (2025) | 3.0 | 29.3 | 53.9 | 11.4 | – | 46.5 |
| Qwen2.5-VL-72B (2025) | 12.0 | 35.2 | 66.9 | 13.0 | – | 47.0 |
| LLaVA-Video-7B (2025d) | 3.1 | 44.6 | 88.7 | 17.7 | – | 48.5 |
| BAGEL-7B-FT (2025a) | 14.7 | 46.3 | 95.5 | 18.7 | 27.0 | 57.2 |
| SPAR-7B (2025a) | **15.3** | – | 90.7 | – | – | 58.1 |
| Oryx-34B (2024) | – | 37.3 | 72.3 | 15.0 | – | – |
| Spatial-MLLM-4B (2025) | 14.8 | 45.0 | 91.8 | 18.4 | – | 55.9 |
| AFFIN-SPACE-4B | 14.9 | 46.4 | 93.8 | 18.7 | 27.2 | 57.6 |
| AFFIN-SPACE-8B | 15.1 | **49.9** | **99.4** | **19.3** | **29.1** | **59.2** |

*Table 3.* **Comparison on Scan2Cap.** AFFIN-SPACE demonstrates strong captioning capabilities, consistently achieving higher scores than prior baselines.

| Methods | Year | B-4 | Rouge | CIDEr | Meteor |
|---|---|---|---|---|---|
| Scan2Cap | 2021 | 23.3 | 44.8 | 39.1 | 22.0 |
| D3Net | 2022 | 30.3 | 51.7 | 46.1 | 24.4 |
| LL3DA | 2024a | 36.8 | – | 65.2 | – |
| Grounded 3D-LLM | 2024b | 35.5 | – | 70.6 | – |
| Chat-Scene | 2024 | 36.3 | 58.1 | 77.1 | 28.0 |
| LLaVA-3D | 2024 | 42.6 | 63.4 | 84.1 | 29.0 |
| Video-3D LLM | 2025c | 41.3 | – | 83.8 | – |
| VG-LLM-4B | 2025b | 40.9 | 62.4 | 78.6 | 28.6 |
| VG-LLM-8B | 2025b | 41.5 | 62.6 | 80.0 | 28.9 |
| AFFIN-SPACE-4B | 2026 | 41.3 | 63.4 | 84.4 | 28.9 |
| AFFIN-SPACE-8B | 2026 | **44.2** | **65.9** | **87.4** | **31.0** |

*Table 4.* **Comparison on EmbodiedScan.** AFFIN-SPACE consistently outperforms VG-LLM across both 4-frame and 6-frame settings.

| Model | $P_{25}$ | $R_{25}$ | $F1_{25}$ |
|---|---|---|---|
| *4-Frame Setting* | | | |
| Qwen2.5-VL-3B | 32.6 | 27.9 | 30.0 |
| VG-LLM-4B | 41.7 (+9.1) | 35.7 (+7.8) | 38.2 (+8.2) |
| AFFIN-SPACE-4B | 43.1 (+10.5) | 38.5 (+10.6) | 41.4 (+11.4) |
| Qwen2.5-VL-7B | 34.6 | 31.0 | 32.5 |
| VG-LLM-8B | 43.4 (+8.8) | 39.6 (+8.6) | 41.2 (+8.7) |
| AFFIN-SPACE-8B | 46.8 (+12.2) | 43.0 (+12.0) | 44.3(+11.8) |
| *6-Frame Setting* | | | |
| Qwen2.5-VL-3B | 27.8 | 24.1 | 25.7 |
| VG-LLM-4B | 39.7 (+11.9) | 34.0 (+9.9) | 36.4 (+10.7) |
| AFFIN-SPACE-4B | 43.0 (+15.2) | 38.1 (+14.0) | 41.8 (+16.1) |
| Qwen2.5-VL-7B | 31.8 | 28.0 | 29.6 |
| VG-LLM-8B | 43.5 (+11.7) | 38.7 (+10.7) | 40.8 (+11.2) |
| AFFIN-SPACE-8B | 47.0 (+15.2) | 42.8 (+14.8) | 44.5 (+14.9) |

reaches a CIDEr score of 87.4, surpassing the previous best LLaVA-3D by a clear margin, and outperforming the video-based competitor Video-3D LLM. Even our smaller AFFIN-SPACE-4B model is highly competitive, achieving a CIDEr of 84.4, which exceeds widely used models like Chat-Scene and matches the performance of significantly larger baselines. The results confirms that our affine-invariant representations effectively capture fine-grained geometric details required for generating dense, accurate scene descriptions. On **EmbodiedScan**, presented in Table 4, AFFIN-SPACE demonstrates superior visual grounding capabilities in both 4-frame and 6-frame settings. Compared to the Qwen2.5-VL backbone, our method yields substantial improvements, proving that standard MLLMs struggle with 3D localization without explicit geometric alignment. Crucially, AFFIN-SPACE consistently outperforms the strong baseline VG-LLM. In the 8B category, we surpass VG-LLM by +3.1 F1 in the 4-frame setting and +3.7 F1 in the 6-frame setting. The performance gains are robust across different frame densities, indicating that our explicit resampling mechanism enables the model to reliably aggregate spatial information from varying numbers of viewpoints for precise detection.

### 4.4. Ablation Study

In this section, we analyze the contribution of key components using the ScanQA and Scan2Cap validation sets.

**Affine Resampling.** We first evaluate the contribution of the Explicit Geometric Resampling module by removing it from the full framework. As shown in Table 5, this leads to a precipitous performance drop across all tasks. Specifically, on ScanQA (8B), the CIDEr score suffers a sharp decline of 8.7, and on Scan2Cap, it drops by 8.0. The results indicates that without explicit spatial alignment, the MLLM struggles

against a broad spectrum of methods. The original Scan2Cap baseline (Chen et al., 2021) and D3Net (Chen et al., 2022), which operate on 3D point clouds. Recent strong baselines that fuse point cloud features with LLMs, including LL3DA (Chen et al., 2024a), Grounded 3D-LLM (Chen et al., 2024b), Chat-Scene (Huang et al., 2024), and LLaVA-3D (Chen et al., 2024a). Video-3D LLM (Zheng et al., 2025c) and VG-LLM (Zheng et al., 2025b), which is similar to our setting, processing multi-view images. For **EmbodiedScan**, we compare our model with the vanilla backbone Qwen2.5-VL (Bai et al., 2025) to demonstrate the gains from our geometric design, and VG-LLM (Zheng et al., 2025b), a state-of-the-art framework for visual grounding in 3D environments.

**Results.** On **Scan2Cap**, reported in Table 3, AFFIN-SPACE achieves state-of-the-art performance across all captioning metrics. It is worth noting that AFFIN-SPACE-8B

*Table 5.* **Effect of Explicit Geometric Resampling.** We remove the affine resampling module to evaluate its contribution. The precipitous drop across all metrics, particularly the sharp decline in CIDEr, indicates that explicit spatial alignment is foundational for the model to handle viewpoint variations.

| Method | B-4 | Rouge | CIDEr | Meteor |
|---|---|---|---|---|
| *ScanQA* | | | | |
| AFFIN-SPACE-8B | 15.1 | 49.9 | 99.4 | 19.3 |
| *w/o* Affine Resampling | 13.9 | 47.1 | 90.7 | 18.3 |
| Δ *Improvement* | (-1.2) | (-2.8) | (-8.7) | (-1.0) |
| AFFIN-SPACE-4B | 14.9 | 46.4 | 93.8 | 18.7 |
| *w/o* Affine Resampling | 13.8 | 44.0 | 86.6 | 17.4 |
| Δ *Improvement* | (-1.1) | (-2.4) | (-7.2) | (-1.3) |
| *Scan2Cap* | | | | |
| AFFIN-SPACE-8B | 44.2 | 65.9 | 87.4 | 31.0 |
| *w/o* Affine Resampling | 38.2 | 59.6 | 76.4 | 28.1 |
| Δ *Improvement* | (-6.0) | (-6.3) | (-11.0) | (-2.9) |
| AFFIN-SPACE-4B | 41.3 | 63.4 | 84.4 | 28.9 |
| *w/o* Affine Resampling | 38.9 | 59.3 | 76.7 | 28.0 |
| Δ *Improvement* | (-2.4) | (-4.1) | (-7.7) | (-0.9) |

to handle geometric variations caused by viewpoint changes, effectively losing its precise spatial reasoning capabilities. Notably, the contrast between the severe drop in sensitivity-prone metrics like CIDEr and the relative stability of Meteor suggests that while the ablated model retains basic semantic understanding, it fails to preserve the geometric fidelity required for accurate spatial grounding.

*Table 6.* **Effect of Affine-Invariant Constraints.** We analyze the effect of removing the orthogonal projection constraints from the latent space. The consistent performance degradation confirms that enforcing strict mathematical invariance is crucial for robustness, preventing the representation from collapsing into view-dependent patterns.

| Method | B-4 | Rouge | CIDEr | Meteor |
|---|---|---|---|---|
| *ScanQA* | | | | |
| AFFIN-SPACE-8B | 15.1 | 49.9 | 99.4 | 19.3 |
| *w/o* Affine-Inv. Constraints | 14.7 | 48.4 | 94.1 | 18.2 |
| Δ *Improvement* | (-0.4) | (-1.5) | (-5.3) | (-1.1) |
| AFFIN-SPACE-4B | 14.9 | 46.4 | 93.8 | 18.7 |
| *w/o* Affine-Inv. Constraints | 14.0 | 44.4 | 90.7 | 17.2 |
| Δ *Improvement* | (-0.9) | (-2.0) | (-3.1) | (-1.5) |
| *Scan2Cap* | | | | |
| AFFIN-SPACE-8B | 41.3 | 63.4 | 84.4 | 28.9 |
| *w/o* Affine-Inv. Constraints | 39.3 | 60.5 | 80.2 | 27.7 |
| Δ *Improvement* | (-2.0) | (-2.9) | (-4.2) | (-1.2) |
| AFFIN-SPACE-4B | 41.3 | 63.4 | 84.4 | 28.9 |
| *w/o* Affine-Inv. Constraints | 39.1 | 60.9 | 78.9 | 27.9 |
| Δ *Improvement* | (-2.2) | (-2.5) | (-5.5) | (-1.0) |

**Affine-Invariant Constraints.** We further analyze the impact of the Affine-Invariant Constraints. The results in Table 6 show that even with resampling enabled, removing these constraints incurs significant performance penalties. This confirms that explicit resampling alone is insufficient.

*Table 7.* **Effect of reconstruction loss coefficient.** The red row denotes the baseline without the reconstruction objective.

| Recon. Loss Coef. | ScanQA | | | | | SQA3D |
|---|---|---|---|---|---|---|
| | B-4 | Rouge | CIDEr | Meteor | EM | EM |
| 0.0 | 14.2 | 45.2 | 92.8 | 18.3 | 26.5 | 55.2 |
| 0.5 | 15.1 | 49.9 | 99.2 | 19.2 | 28.9 | 59.0 |
| 1.0 | 15.1 | 49.9 | 99.4 | 19.3 | 29.1 | 59.2 |
| 2.0 | 15.0 | 50.2 | 89.9 | 19.2 | 29.2 | 58.7 |
| 5.0 | 14.7 | 49.5 | 97.8 | 18.7 | 28.6 | 58.6 |

The strict mathematical invariance enforced by our constraints acts as a critical regularizer. It prevents the latent representation from collapsing into view-dependent patterns, ensuring that the learned features remain robust and geometrically consistent regardless of the input viewpoint.

**Reconstruction Loss** We further analyze the reconstruction loss coefficient $\lambda$ in Table 7, which balances invariance with informativeness. The baseline without reconstruction ($\lambda = 0.0$) yields the lowest performance, suggesting that enforcing invariance alone leads to representation collapse where geometric details are discarded. Increasing $\lambda$ consistently improves performance, peaking at $\lambda = 1.0$. The results confirm the auxiliary reconstruction objective acts as a critical regularizer, ensuring the invariant representations retain sufficient information to recover the underlying 3D structure. However, excessively strong regularization ($\lambda > 1.0$) hampers the main task learning; thus, we adopt $\lambda = 1.0$ as the optimal setting.

## 5. Conclusion

In this work, we addressed the geometric fragility inherent in current MLLMs, identifying that standard visual representations remain entangled with the view-dependent pixel grid and degrade rapidly under affine perturbations. To overcome this, we presented **AFFIN-SPACE**, a framework that enforces strict affine invariance by explicitly decoupling intrinsic 3D structure from extrinsic camera pose. Our approach introduces two core mechanisms: (i) explicit geometric resampling, which aligns 3D inputs to a canonical state using decomposed affine quantities, and (ii) affine-invariant constraints via orthogonal projection, which mathematically strips away pose-dependent noise from the latent space. Empirical results across VSI-Bench, ScanQA, SQA3D, Scan2Cap, and EmbodiedScan demonstrate that AFFIN-SPACE not only achieves state-of-the-art performance but also exhibits superior stability against viewpoint changes. Our findings confirm that explicitly modeling geometric invariance, rather than relying on implicit feature fusion, is a critical step toward developing robust and reliable spatially intelligent systems.

## Acknowledgements

We sincerely thank the anonymous reviewers and chairs for their efforts and constructive suggestions, which have greatly helped us improve the manuscript. This work is supported in part by the National Natural Science Foundation of China under grants 62536003, 62521006, and 624B2088, and by the project of Peng Cheng Laboratory (PCL2025A14).

## Impact Statement

The introduction of AFFIN-SPACE can enhance the robustness of 3D MLLMs by enabling them to maintain consistent spatial understanding across varying viewpoints and viewing distances, a capability critical for the deployment of reliable embodied AI and autonomous systems. By explicitly modeling affine invariance, our framework addresses the "geometric fragility" of current models, thereby reducing the risk of failure in safety-critical applications—such as search-and-rescue robotics or assistive devices for the visually impaired—where maintaining a stable understanding of the environment despite camera motion is paramount. While we acknowledge that improved object re-identification under geometric distortion could theoretically strengthen surveillance capabilities, we emphasize that the primary utility of our method lies in eliminating the "viewpoint dependence" that currently limits the safe generalization of vision systems. Consequently, this work encourages the development of more trustworthy AI agents that ground their reasoning in stable physical properties rather than transient visual patterns, promoting safer human-AI interaction in dynamic, real-world environments.

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
