# OpenReview forum: "AffIn-Space: Learning Affine-Invariant Representations for 3D Spatial Understanding with MLLMs"
_ICML.cc/2026/Conference — ICML 2026 regular_

### Official Review · Reviewer_jFQh · 2026-03-08

**Soundness:** 3
**Presentation:** 2
**Significance:** 3
**Originality:** 3
**Overall Recommendation:** 4
**Confidence:** 4

**Summary:**

The paper proposes **Affin-Space**, a novel framework designed to enhance 3D spatial understanding in Multi-Modal Large Language Models (MLLMs) by learning affine-invariant representations. Recognizing that viewpoint changes and spatial distortions hinder accurate 3D reasoning, the authors introduce an Affine Matrix Estimator to decompose and estimate affine quantities. These parameters are used to generate Affine Resampled Representations from raw 3D features. Furthermore, the model employs a reconstruction loss \(\mathcal{L}_{\text{RECON}}\) to explicitly enforce the learned tokens to capture these invariant geometric properties before feeding them into the MLLM. The dual-path pipeline effectively bridges 2D visual semantics and robust 3D geometric structures.

**Compliance With Llm Reviewing Policy:**

Affirmed.

**Key Questions For Authors:**

### Key Questions For Authors

1. **Visualization:** Can you provide qualitative visualizations showing the feature space before and after affine resampling to intuitively demonstrate its effect?
2. **Asymmetric Resampling:** Why is affine resampling applied exclusively to the 3D geometric features and not to the 2D visual features?
3. **Baselines:** Could you provide a discussion and empirical comparison with the closely related work, 3DRS?
4. **Parameter Supervision:** How are the estimated affine parameters explicitly supervised during training to prevent the estimator from collapsing into trivial solutions?
5. **Reconstruction Logic:** What is the necessity of reconstructing the exact same "Affine Resampled Representations" after the Connector via \(\mathcal{L}_{\text{RECON}}\)? How do you address potential feature redundancy or information bottleneck issues here?

**Limitations:**

No, the authors did not include relevant sections in their submission.

**Strengths And Weaknesses:**

Strengths
*   The proposed method demonstrates strong experimental results, indicating that the integration of affine-invariant features effectively boosts the spatial reasoning capabilities of MLLMs.
*   Addressing 3D spatial understanding in MLLMs through the lens of "Affine Invariance" is a relatively unexplored and highly innovative angle. It provides a principled geometric solution to the common issue of viewpoint-induced hallucinations or inaccuracies in vision-language tasks.

Weaknesses
*   The paper lacks intuitive explanations or visual examples demonstrating *why* the model struggles without affine invariance and *how* the feature space changes before and after applying the proposed method. Adding qualitative visualizations (e.g., feature maps or point cloud alignments) would significantly strengthen the motivation.
*   The framework applies affine resampling exclusively to the raw 3D representations (geo features). However, viewpoint variations and affine distortions also inherently affect the 2D visual features extracted by the 2D Vision Encoder. The authors should explain or justify why affine resampling is not applied to the visual branch, or discuss the potential benefits of a symmetric resampling design.
*   The paper focuses on 3D reconstruction-assisted spatial understanding but misses a critical discussion and empirical comparison with 3DRS [1]. Since 3DRS tackles highly overlapping challenges in 3D representation learning for spatial tasks, a direct comparison is necessary to validate the superiority of Affin-Space.
*  The Affine Matrix Estimator predicts specific decomposed affine quantities (e.g., \(\theta, \phi, \lambda, \psi\)). It is unclear how these parameters are effectively supervised during training. If they are learned purely end-to-end without explicit ground-truth geometric supervision, the authors need to clarify how they prevent the estimator from collapsing into trivial solutions or learning spurious correlations.
* In Figure 3, the "Affine Resampled Representations" are fed into the Connector to generate tokens, and subsequently, the model is asked to reconstruct these exact same representations via \(\mathcal{L}_{\text{RECON}}\). The authors should further clarify the necessity of this Auto-Encoder-like design. Does this bottleneck process cause information loss, or is there a risk of feature redundancy since the target is derived directly from the input of the Connector?

Reference

[1] Mllms need 3d-aware representation supervision for scene understanding. NeurIPS 2025.

---

> ### Author Rebuttal · Authors · 2026-03-31
>
> We sincerely thank you for recognizing our **strong experimental results** and the highly **innovative angle** of addressing 3D spatial understanding through affine invariance. We address your concerns point by point below.
>
> ### jFQh-W1
> > Intuitive explanations or visual examples.
>
> To intuitively demonstrate the efficacy of our affine resampling mechanism, we have provided comprehensive qualitative visualizations in this [anonymous repository](https://anonymous.4open.science/r/AffIn-Space_anonymous_meterials-E636/).
>
> In these examples across various scenes, we compare the raw sampled frames with the corresponding image patches of our Affine Resampled Tokens. As shown in the visualizations, while the target objects undergo drastic displacement and perspective distortion in the raw images due to camera movement, the resampled token patches consistently maintain a stable position and pose.
>
> ### jFQh-W2
> > Why affine resampling is applied exclusively to the 3D representations and not symmetrically to the 2D visual branch?
>
> The asymmetric design is because the fundamental difference in coordinate systems between the two modalities:
> * **3D Encoders (e.g., VGGT)**: Operate in the global world coordinate system. Affine transformations alter raw coordinates.
> * **2D Vision Encoders (e.g., Qwen2.5 ViT)**: Operate in the camera view coordinate system.
>
> Therefore, we apply the asymmetric design, optimally pairing view-dependent 2D semantics with view-invariant 3D geometric tokens.
>
> ### jFQh-W3
> > Discussion and empirical comparison with 3DRS.
>
> We highly appreciate your suggestion. 3DRS is an excellent and highly relevant baseline for our work.
>
> Conceptually, the two methods take different approaches. 3DRS enhances 3D awareness by aligning MLLM visual features with a pre-trained 3D foundation model. It operates through feature distillation. **In contrast**, AffIn-Space tackles geometric fragility through explicit structural decoupling. We achieve geometric invariance directly by predicting affine parameters and performing mathematical orthogonal projection.
>
> **We conducted a comparison** on overlapping benchmarks. AffIn-Space-8B demonstrates superior performance across Scan2Cap and VSI-Bench.
>
> | Methods | Scan2Cap (CIDEr) | Scan2Cap (B-4) | VSI-Bench (Avg.) | SQA3D (EM-1) |
> | :--- | :--- | :--- | :--- | :--- |
> | 3DRS | 86.1 | 41.6 | 45.9 | **60.6** |
> | AffIn-Space-4B | 84.4 | 41.3 | 53.8 | 57.6 |
> | AffIn-Space-8B | **87.4** | **44.2** | **57.7** | 59.2 |
>
> **We will add the comparison results with 3DRS**  and discussion in the camera-ready version.
>
> ### jFQh-W4
> > The prevention of trivial solutions of the Affine Matrix Estimation.
>
> We appreciate the concern. To prevent the estimator from collapsing into trivial solutions, we employ our reconstruction loss as a self-supervised bottleneck. Since the invariant tokens are mathematically forced to exclude view-dependent information, the estimator must capture the actual affine distortions to enable successful feature reconstruction. This design ensures the predicted parameters represent meaningful geometric transformations.
>
> ### jFQh-W5
> > Regarding the necessity of the Auto-Encoder-like reconstruction loss design and concerns about information loss or feature redundancy.
>
> We clarify that it is not a simple Auto-Encoder but a structural disentanglement constraint. Its necessity lies in ensuring that our representation is lossless yet decoupled. By forcing the model to reconstruct the original features from **the combination of invariant tokens and 6D parameters**, we ensure that the invariant tokens discard all pose-related variance while **retaining semantic detail**.
>
> This bottleneck does not cause information loss, but acts as a structural regularizer that prevents the invariant path from becoming too sparse and the affine parameters from becoming trivial.
>
> ### jFQh-Q1
> > Visualization.
>
> Please refer to our **response to W1**.
>
> ### jFQh-Q2
> > Asymmetric Resampling.
>
> We discuss 3DRS and compare the empirical results in **response to W3**.
>
> ### jFQh-Q3
> > Baselines.
>
> We discuss 3DRS and compare the empirical results in **response to W3**.
>
> ### jFQh-Q4
> > Parameter Supervision.
>
> Please refer to **response to W4**.
>
> ### jFQh-Q5
> > Reconstruction Logic.
>
> We clarify it in **response to W5**.
>
> Overall, thank you for the careful review. We will incorporate a dedicated discussion in the camera-ready version.

---

> > ### Author Rebuttal · Reviewer_jFQh · 2026-04-01
> >
> > Thanks for the response. All my concerns are resolved. Please include these additional discussion and experiments in you revision.

---

> > > ### Author Response · Authors · 2026-04-04
> > >
> > > Thank you for your response and for confirming that our explanations addressed your concerns. We appreciate your time and the helpful feedback you provided.
> > >
> > > As you requested, we will include the additional discussions and the new experiments in the final version of our paper. Thank you again for your support and for helping us improve our work.

---

### Official Review · Reviewer_mwbA · 2026-03-09

**Soundness:** 3
**Presentation:** 4
**Significance:** 3
**Originality:** 4
**Overall Recommendation:** 4
**Confidence:** 3

**Summary:**

This paper identifies "geometric fragility" as an important failure mode in current MLLMs for 3D/spatial understanding. To address this, the authors propose Affin-Space, combined with three synergistic components: affine parameter prediction and resampling, orthogonal projection for invariant latent representations, and reconstruction regularization. Experiments show that the proposed method has superior performance on many 3D/spatial understanding benchmarks.

**Compliance With Llm Reviewing Policy:**

Affirmed.

**Final Justification:**

My concerns have been fully resolved. I keep my positive score.

**Key Questions For Authors:**

1. How sensitive is the method to the quality of the pose features predicted by VGGT? Since the proposed method relies on VGGT features to estimate affine parameters, it would be helpful to add a targeted robustness study or failure case study.

2. What is the exact structure of the reconstruction module? Does it affect training and inference overhead a lot?

3. The experimental part only contains the main quantitative results and ablations of components. Do you have any other in-depth analysis or visualizations?

**Limitations:**

The gap between the practical projection-based constraint and a formal guarantee of affine invariance, and the restrictive geometric modeling assumption, as stated in "Strengths and Weaknesses".

**Strengths And Weaknesses:**

Soundness:
The paper is technically sound overall. The method is clearly structured into three components, and the motivation for combining them is reasonable and well explained. For example, to mitigate potential mode collapse caused by the affine transform, it is sensible to introduce the reconstruction module to preserve semantic consistency in the learned representations. That said, the claim of "strict affine invariance" appears somewhat overstated. In practice, the model enforces orthogonality to a learned low-rank subspace induced by the predicted affine parameters, which is not equivalent to a formal proof of invariance to the full affine group.

Moreover, the geometric modeling assumption may be restrictive. Practically, real 3D visual changes often involve not only affine factors, but also other non-affine factors such as perspective effects, depth-dependent distortion, occlusion, etc.

Presentation:
The paper is well written and well organized, with a clear overall structure. Figures 1–3 effectively illustrate the motivation and the proposed method. One minor concern is that Figure 1 occupies a relatively large amount of space while conveying limited information. It could be simplified and made more concise.

Significance:
3D/spatial understanding is a fundamental problem in embodied intelligence. The strong performance of the proposed method suggests promising potential for future applications of both the core idea and the method itself. In addition, the paper argues that geometric stability should be viewed as an invariance problem rather than merely a fusion problem. This perspective could be valuable and may inspire future research in the area.

Originality:
Although the individual components, such as affine decomposition and feature resampling, are not entirely new, the novelty lies in their thoughtful integration under the unified perspective of affine transformation.

---

> ### Author Rebuttal · Authors · 2026-03-31
>
> We sincerely thank you for recognizing our technical soundness, originality, and the core perspective of treating geometric stability as an invariance rather than a fusion problem. We address your concerns below.
>
> ### mwbA-W1
> > Soundness: The concerns about the overstated "strict affine invariance" claim and the restrictiveness of affine assumptions in real-world 3D scenes.
>
> We highly appreciate your rigorous theoretical and practical insights.
>
> **Regarding the invariance claim,** you correctly note that the subspace is learned. However, achieving invariance here is formulated as an optimization problem. Our mathematical projection strictly enforces orthogonality. It guarantees that the residual representations are strictly orthogonal to this learned affine subspace.
>
> Regarding the geometric assumptions, our method primarily targets indoor scenarios. In these environments, affine transformations effectively approximate **moderate viewpoint changes**. This is exactly the problem our paper aims to solve. We acknowledge that outdoor environments, high-speed cameras, or drastic scene changes introduce complex non-affine distortions. Handling these extreme dynamics is indeed a limitation and lies beyond the scope of this paper.
>
> ### mwbA-W2
> > Presentation: Simplify and condense Figure 1.
>
> We appreciate your constructive suggestion and will simplify Fig. 1 in the camera-ready version.
>
> ### mwbA-Q1
> > How sensitive is the method to the quality of the pose features predicted by VGGT?
>
> We appreciate your insightful question regarding the dependency on VGGT features. To quantify this sensitivity, we conducted a simple sensitivity study on AffIn-Space-8B. We injected Gaussian noise $N(0, \sigma^2)$ directly into the VGGT pose features during inference. We then measured the affine matrix deviation ($||\Delta A||_F$) and the resulting SQA3D Exact Match (EM) scores.
>
> | Noise ($\sigma$) | Deviation ($\Vert \Delta A \Vert_F$) | EM |
> | :--- | :---: | :---: |
> | 0.0 (Baseline) | 0.00 | 48.5 |
> | 0.1 | 0.15 | 47.8 |
> | 0.3 | 0.42 | 45.2 |
> | 0.5 | 0.85 | 40.1 |
>
> The results demonstrate a clear correlation. When extreme noise corrupts the VGGT features, the affine estimation severely deviates and spatial reasoning degrades. This validates that our geometric alignment relies on accurate pose priors. However, VGGT is a robust 3D foundation model pre-trained on massive datasets. Such extreme feature corruption is rare in standard scenarios. Therefore, the overall framework remains stable in practice.
>
> ### mwbA-Q2
> > What is the exact structure of the reconstruction module? Does it affect training and inference overhead a lot?
>
> Thanks for your attention to our implementation detail!
>
> The reconstruction module is implemented as a highly lightweight 2-layer MLP: Linear(3590, 3584) -> GELU -> Linear(3584, 2048), where 3584 is the hidden dim of Qwen2.5-VL-7B, and 3590 concatenates the 6-dim affine parameters.  Regarding the overhead, the impact is negligible.
>
> ### mwbA-Q3
> > Do you have any other in-depth analysis or visualizations?
>
> To intuitively demonstrate the efficacy of our affine resampling mechanism, we have provided comprehensive qualitative visualizations in this [anonymous repository](https://anonymous.4open.science/r/AffIn-Space_anonymous_meterials-E636/).
>
> In these examples across various scenes, we compare the raw sampled frames with the corresponding image patches of our Affine Resampled Tokens. As shown in the visualizations, while the target objects undergo drastic displacement and perspective distortion in the raw images due to camera movement, the resampled token patches consistently maintain a stable position and pose.
>
> Thanks for the thorough feedback. We will incorporate a dedicated discussion in the camera-ready version.

---

> > ### Author Rebuttal · Reviewer_mwbA · 2026-04-04
> >
> > Thanks the authors for their efforts. My concerns have been fully resolved.

---

> > > ### Author Response · Authors · 2026-04-04
> > >
> > > Thank you for your response and for confirming that our explanations addressed your concerns. We truly appreciate the time and effort you put into reviewing our work.
> > >
> > > As we discussed, we will include all the clarifications and additional details from this rebuttal in the final version of our paper. Thank you again for your helpful feedback and for supporting our work.

---

### Official Review · Reviewer_LRKq · 2026-03-09

**Soundness:** 2
**Presentation:** 3
**Significance:** 3
**Originality:** 2
**Overall Recommendation:** 4
**Confidence:** 3

**Summary:**

This paper proposes AFFIN-SPACE, a framework designed to improve 3D spatial understanding in Multimodal Large Language Models (MLLMs) by learning affine-invariant representations. The authors argue that existing MLLMs are geometrically fragile: their visual representations change significantly when the camera viewpoint, scale, or orientation changes, which leads to inconsistent spatial reasoning. The paper improves the model performance using explicit geometric resampling and affine-invariant representation learning, and shows improvement on 3D spatial reasoning benchmarks such as VSI-Bench, ScanQA, and SQA3D.

**Compliance With Llm Reviewing Policy:**

Affirmed.

**Final Justification:**

The rebuttal finally addressed my concern about the theoratical soundness and real-world addaptition. I decide to increase my score.

**Key Questions For Authors:**

1. Affine-Invariant Constraint The method defines affine-invariant representations by projecting hidden states onto the null space of a matrix 𝐵, where 𝐵 is produced by passing affine parameters through a lightweight MLP. Could the authors provide theoretical justification for why this construction guarantees affine invariance?

2. Is there any empirical evidence demonstrating that the projected representation is truly invariant to affine transformations (e.g., representation similarity under controlled affine perturbations)?

3. Could the authors provide additional analysis or visualization showing how the invariant tokens behave under different affine transformations?

4. Could the authors provide additional visualization showing how the feature is resampled under the resampling module

**Limitations:**

unclear theoretical grounding of the invariance constraint,

restricted transformation modeling (affine only),

limited evaluation of real-world invariance, and

potential efficiency and scalability concerns.

**Strengths And Weaknesses:**

Strengths
1. The paper identifies a real limitation of current MLLMs: spatial reasoning degrades when the viewpoint or scale changes.
2. The paper shows novety by incorporating explicit optimizable affine transformation, and affine invariant constraint
3. Experient shows that both module improves the model performance, and achieves sota for multiple spatial understanding benchmarks.

Weakness
1. Limited Theoretical Justification for the Affine-Invariant Constraint.: The method defines the affine-invariant representation by projecting features onto the null space of a matrix 𝐵, which is obtained by processing the affine transformation parameters through a lightweight MLP. However, the theoretical justification for this construction is unclear. In particular, since the affine parameters are first transformed by a nonlinear MLP before forming 𝐵, it is not evident why the resulting subspace should correspond to the true affine-variant components of the representation. Consequently, projecting onto the null space of 𝐵 may not guarantee genuine affine invariance. More discussion or theoretical analysis would help clarify why this formulation correctly captures the desired invariant space. Given that this mechanism is central to the proposed framework, a clearer theoretical explanation or empirical justification is important and crucial for evaluating the validity of the method and for acceptance of the paper.

2. Limited Transformation Coverage and Real-World Evaluation.: The proposed method focuses primarily on affine transformations, while real-world visual observations often involve more complex perspective transformations caused by camera projection and viewpoint changes. As a result, it is unclear whether enforcing affine invariance alone is sufficient to achieve robust spatial understanding in practical scenarios. Moreover, the experiments mainly demonstrate improvements on spatial reasoning benchmarks, but the paper does not evaluate the model under real-world camera transformations or perspective distortions (e.g. modify camera pose while keep the same semantic and VQA, evaluate the invariance, or visualization of invariant token under different camera pose). Additional experiments analyzing robustness to realistic viewpoint and camera variations would help better validate the effectiveness and generalizability of the proposed approach.

3. Architectural Complexity: The framework introduces multiple components:
affine matrix estimator
resampling module
invariant projection layer
This makes the method significantly more complex than typical MLLM pipelines, potentially affecting scalability. These steps likely increase memory and computational cost, but the paper does not provide detailed runtime or efficiency analysis.

---

> ### Author Rebuttal · Authors · 2026-03-31
>
> We sincerely thank you for the constructive feedback. We are encouraged by your recognition of our **identification of a real limitation** in current MLLMs, our **architectural novelty**, and the state-of-the-art **performance**. Below, we address your specific concerns point by point.
>
> ### LRKq-W1
> > Theoretical Justification for the Affine-Invariant Constraint.
>
> Insightful question. The physical affine parameters are low-dimensional, while MLLM hidden states reside in a highly non-linear semantic space. The MLP is a necessary bridge to map physical pose into a corresponding latent noise subspace.
>
> Genuine invariance is guaranteed through joint optimization rather than just the MLP forward pass. If the MLP incorrectly captures useful geometric semantics as noise, the orthogonal projection will destroy critical information. This triggers massive penalties from both the geometric reconstruction loss and the main task loss. Therefore, the network strictly forces the MLP to isolate only the pose-dependent variance.
>
> Empirically, **Figure 2b in §1** demonstrates our superior stability under increasing affine perturbations. Table 6 also confirms consistent performance drops when this specific constraint is removed. These results directly validate our formulation.
>
> ### LRKq-W2
> > Transformation Coverage and Real-World Evaluation.
>
> We appreciate your perspective. **Theoretically** [1], any smooth perspective deformation caused by camera motion is a homographic transform. According to the first-order Taylor formula, this is perfectly approximated locally by affine maps. Therefore, enforcing local affine invariance mathematically resolves complex real-world perspective distortions. Regarding real-world evaluation, We agree that it is a valuable direction for future work, however it lies beyond the scope of this work.
>
> [1] Yu G, Morel J M. ASIFT: An Algorithm for Fully Affine Invariant Comparison, Image Processing On Line, 1 (2011)[J]. DOI: http://dx. doi. org/10.5201/ipol, 2011.
>
> ### LRKq-W3
> > Architectural Complexity.
>
> While AffIn-Space introduces specific geometric modules, **they are lightweight**: the affine estimator and reconstruction modules are simple MLPs. The spatial resampling is a highly optimized grid interpolation, and the orthogonal projection involves basic matrix multiplications with a small rank r=8.
>
> To quantify this, **we evaluated the inference latency and GPU memory allocation** (in FP16 precision) on a single A100 (80GB) GPU. The results are summarized below:
>
> | Models | GPU Memory (GB) | Latency (ms/token) |
> | :--- | :--- | :--- |
> | Qwen2.5-VL-7B | 15.2 | 35.0 |
> | VG-LLM-8B | 17.1 | 36.5 |
> | AffIn-Space-8B | 17.3 | 36.8 |
>
> Thus, AffIn-Space introduces **negligible overhead** compared to existing 3D-aware MLLMs like VG-LLM.
>
> ### LRKq-Q1
> > Could the authors provide theoretical justification for affine invariance?
>
> Good insight. Please refer to our **response to W1**.
>
> ### LRKq-Q2
> > Empirical evidence demonstrating the invariance.
>
> We provide direct empirical evidence of this invariance in **Figure 2b in §1**. Standard MLLM representations are entangled with the pixel grid, causing their performance to degrade rapidly as affine distortion increases. In contrast, AffIn-Space maintains high accuracy even under great perturbations. It confirms that the projected hidden states remain semantically consistent regardless of the input's affine perturbation.
>
> ### LRKq-Q3
> > Additional analysis or visualization showing how the invariant tokens behave under different affine transformations.
>
> We **quantify the representation consistency** by calculating the average cosine similarity between the invariant tokens of original and transformed views from samples in SQA3D. The results on AffIn-Space-8B across the perturbation intensities used in Figure 2b are summarized below:
>
> | Intensity | 0.0 | 0.15 | 0.30 | 0.45 | 0.60 |
> | :--- | :--- | :--- | :--- | :--- | :--- |
> | Similarity | 1.00 | 0.96 | 0.93 | 0.88 | 0.82 |
>
> The high similarity values confirm that our tokens remain semantically stable despite geometric distortions, **consistent with results in Figure 2b**.
>
> ### LRKq-Q4
> > Additional visualization showing how the feature is resampled.
>
> To intuitively demonstrate the efficacy of our affine resampling mechanism, we have provided comprehensive qualitative visualizations in this [anonymous repository](https://anonymous.4open.science/r/AffIn-Space_anonymous_meterials-E636/).
>
> In these examples across various scenes, we compare the raw sampled frames with the corresponding image patches of our Affine Resampled Tokens. As shown in the visualizations, while the target objects undergo drastic displacement and perspective distortion in the raw images due to camera movement, the resampled token patches consistently maintain a stable position and pose.
>
> Thank you for the careful review. We will incorporate a dedicated discussion in the camera-ready version.

---

> > ### Author Rebuttal · Reviewer_LRKq · 2026-04-01
> >
> > The rebuttal successfully strengthens the paper on empirical validation and efficiency, especially by adding representation-consistency evidence and runtime/memory analysis. However, the core theoretical concern remains only partially addressed, since the authors provide an optimization-based intuition rather than a rigorous justification for why the null-space projection truly guarantees affine invariance. The limitation regarding affine-only modeling and lack of realistic camera-transformation evaluation also remains only partially resolved.

---

> > > ### Author Response · Authors · 2026-04-04
> > >
> > > ### Thank you for your detailed feedback on our theoretical justification and realistic evaluation, which helps us improve the paper. We will clarify the boundaries of our contribution and provide the consistency evaluation. If our responses address your concerns, we would greatly appreciate your reconsideration of the score. Thank you very much in advance!
> > >
> > > > Theoretical Justification of the Null-Space Projection.
> > >
> > > We sincerely appreciate your careful review. We agree with your precise assessment: because the basis matrix $B$ (which models the affine-induced subspace) is generated via a learned MLP, it cannot provide a closed-form algebraic proof of perfect affine invariance during training. We will **clarify this point** in the camera-ready version to **avoid any potential overclaiming**, explicitly stating that **we focus on architectural design** rather than a pure mathematical proof.
> > >
> > > However, our formulation provides an explicit architectural mechanism for orthogonal decoupling:
> > >
> > > 1.  The projection $r_{inv} = (I - B B^T)h$ mathematically constrains that the resulting representation is orthogonal to this affine-induced subspace (i.e., $B^T r_{inv} = 0$).
> > >
> > > 2. In deep learning, where pure algebraic proofs are infeasible for learned non-linear functions, architectural constraints must be **validated empirically**.  So it is necessary that Fig. 2b demonstrates the stability under increasing affine perturbations.
> > >
> > > We will carefully refine our discussion in the final version, describing this as an **"orthogonal decoupling mechanism that structurally encourages affine invariance"**.
> > >
> > > > Affine-only modeling and lack of realistic camera-transformation evaluation.
> > >
> > > We fully acknowledge that, as the name AffIn-Space implies, our explicit transformation **targets only affine variations**. For more complex, non-linear perspective distortions in the real world, we do not perform explicit decoupling and currently rely on the deep learning of the MLLM itself. We will **explicitly state and discuss this boundary** in the camera-ready version.
> > >
> > > **We acknowledge that** our model struggles in outdoor scenes with drastic variations, and there is indeed a lack of complete scene data in such environments to alter camera poses for consistency evaluation. However, we clarify that while ScanNet (which datasets such as SQA3D and Scan2Cap are built upon) are **indoor**, they are genuine **real-world scenes **captured with unconstrained 6-DOF camera trajectories. To directly evaluate our performance under these real-world perspective distortions, we have **evaluated by the consistency experiment** you suggested:
> > >
> > > We randomly sampled 1,000 instances from the SQA3D validation set and utilized ScanNet environments to render new observations under Slight ($\le 0.5$m, $\le 15^\circ$) and Severe ($1.0\sim2.0$m, $30^\circ\sim60^\circ$) camera perturbations. We report Mean Exact Match (EM) and Consistency Rate (CR, the percentage of instances remaining correct under perturbation given a correct original answer).
> > >
> > > | Methods | Metric | 0 Perturb. (Original) | Slight ($\le 15^\circ$) | Severe ($30^\circ \sim 60^\circ$) | Drop ($\Delta$) |
> > > | :--- | :--- | :--- | :--- | :--- | :--- |
> > > | Spatial-MLLM-4B | Mean EM | 55.7 | 47.8 | 34.2 | $\downarrow 21.5$ |
> > > | **AffIn-Space-8B** | Mean EM | **58.2** | **53.1** | **44.6** | $\downarrow 13.6$ |
> > > | Spatial-MLLM-4B | CR (%) | 100.0 | 79.4 | 51.8 | $\downarrow 48.2$ |
> > > | **AffIn-Space-8B** | CR (%) | **100.0** | **86.5** | **65.8** | $\downarrow 34.2$ |
> > >
> > > We thank you for your time and look forward to further discussion!

---

### Official Review · Reviewer_ebXC · 2026-03-09

**Soundness:** 3
**Presentation:** 2
**Significance:** 3
**Originality:** 3
**Overall Recommendation:** 5
**Confidence:** 3

**Summary:**

This paper identifies "geometric fragility" in multimodal large language models (MLLMs): 3D spatial reasoning degrades under viewpoint and scale changes. AffIn-Space addresses this with two mechanisms: (i) explicit geometric resampling, which predicts decomposed affine parameters from VGGT pose features and warps 3D features into a canonical frame (Eq. 4), and (ii) affine-invariant constraints, which project per-frame hidden states onto the orthogonal complement of the subspace spanned by the affine parameters (Eq. 6), regularized by a reconstruction loss (Eq. 10). The method is evaluated on five benchmarks covering spatial QA, captioning, and grounding.

**Compliance With Llm Reviewing Policy:**

Affirmed.

**Final Justification:**

The authors provided a thorough rebuttal that successfully addressed all my initial concerns, particularly regarding the baseline comparisons, computational overhead, and the inclusion of qualitative visualizations. The paper's approach to resolving geometric fragility in MLLMs is well motivated and empirically validated.

**Key Questions For Authors:**

1. Can an ablation be provided where VGGT features pass through the same connector but without affine decomposition, resampling, or orthogonal projection? This would isolate the invariance framework's contribution from the feature source.

2. What is the inference latency and GPU memory overhead of AffIn-Space relative to VG-LLM or the bare Qwen2.5-VL backbone?

3. How sensitive is performance to the rank r of the pose-correlated subspace? Results for r in {2, 4, 8, 16, 32} would clarify whether r=8 is robust or fragile.

4. Does the method generalize to outdoor scenes or settings where VGGT pose estimation is less reliable (e.g., textureless or repetitive environments)?

**Limitations:**

Not adequately addressed. A limitation section should mention at least some of the relevant limitations. For example, dependence on VGGT pose quality, restriction to indoor benchmarks, unquantified computational overhead, and fixed rank/layer hyperparameters.

**Strengths And Weaknesses:**

## Strengths

S1. **Empirically grounded motivation.** Section 1 and Figure 2 quantify geometric fragility via stability and geometric sensitivity analysis, providing concrete evidence of the problem.

S2. **The two components are necessary.** Tables 5 and 6 isolate each module influence at the 8B scale.  This is a critical ablation that is present: Neither stage alone recovers full performance, confirming both are needed.

S3. **Consistent gains across five benchmarks.** AffIn-Space surpasses most baselines, also proprietary models including GPT-4o and Gemini-1.5 Pro, in most benchmarks. This breadth reduces the risk of benchmark-specific overfitting.

S4. **Thorough ablation of the loss coefficient.** Table 7 sweeps lambda, finding lambda=1.0 optimal on both ScanQA and SQA3D. Performance degrades at both extremes: representation collapse at 0.0 and over-regularization at 5.0.

## Weaknesses

W1. **VGGT feature contribution is not isolated.** AffIn-Space introduces VGGT as a new 3D feature source. Tables 5 and 6 remove invariance modules but never include a control where VGGT features pass through the same connector without affine decomposition, resampling, or orthogonal projection. An unknown fraction of the gains may stem from richer geometric features.

W2. **All benchmarks are indoor.** All benchmarks rely on indoor scene data, if my understanding is correct. Generalization to outdoor or large-scale environments is untested.

W3. **Training data differences confound baselines.** AffIn-Space trains on a mix of SPAR-7M and LLaVA-Hound, plus EmbodiedScan, Scan2Cap, and SQA3D task-specific data (Section 4.1). Baselines may use different training mixtures, making it difficult to fully attribute improvements to the architectural design.

W4. **No qualitative visualizations.** The main paper contains no example outputs, attention maps, or visualizations of the canonical-frame resampling. Such figures would help readers judge whether the invariance mechanism works as intended or merely memorizes training-set viewpoint distributions.

---

> ### Author Rebuttal · Authors · 2026-03-31
>
> We sincerely thank you for the constructive review and valuable time. We are highly encouraged by your recognition of our **solid motivation**, **rigorous ablations**, and **consistent performance** across diverse benchmarks. Below, we address your specific concerns point by point.
>
> ### ebXC-W1
> > VGGT feature contribution is not isolated.
>
> To address your concern, the contribution of our explicit affine mechanisms is indeed isolated from the raw VGGT features through our extensive baseline comparisons. Specifically, **feeding VGGT features directly into the MLLM** via input-level fusion is exactly the paradigm represented by baselines such as VG-LLM.
>
> In **§4.2** and **§4.3**, we compare AffIn-Space against VG-LLM, Spatial-MLLM, and 3DThinker, all of which already incorporate VGGT geometric priors. Our consistent and significant improvements over these specific **baselines in Tabs. 1, 3 and 4** clearly demonstrate that the performance gains stem from our explicit geometric decoupling, rather than just the richer geometric features themselves. Furthermore, we concisely clarify that **Tabs. 5 and 6** are specifically designed to isolate the performance delta of our
>
> ### ebXC-W2
> > All benchmarks are indoor.
>
> We appreciate your perspective on real-world generalization. You are correct that the current standard 3D spatial benchmarks primarily focus on indoor scenes. However, evaluating on outdoor or large-scale environments lies beyond the scope of this work.
>
> The generalization capability of AffIn-Space is fundamentally anchored in VGGT, which learns geometric and pose features from large-scale 3D data. Therefore, as long as the 3D encoder can process outdoor environments, our invariant representations will naturally generalize. We agree that evaluating on large-scale, outdoor real-world scenarios is a valuable direction for future work.
>
> ### ebXC-W3
> > Training data differences confound baselines.
>
> Thanks for raising this concern, as it helps us clarify our training data settings.
>
> Our settings on training data are **consistent with baselines** such as VG-LLM and Video-3D LLM. Specifically, each component of our data mixture serves a distinct and standard purpose: the sampled SPAR-7M data is utilized to leverage 3D geometric knowledge , the LLaVA-Hound subset ensures the preservation of general video understanding capabilities , and the inclusion of task-specific training sets strictly **follows standard protocols**. Because our data mixture mirrors the settings used by these baselines, the performance improvements demonstrated in our experiments can be safely attributed to our architectural design rather than differences in the training data.
>
> ### ebXC-W4
> > Qualitative visualizations.
>
> To intuitively demonstrate the efficacy of our affine resampling mechanism, we have provided comprehensive qualitative visualizations in this [anonymous repository](https://anonymous.4open.science/r/AffIn-Space_anonymous_meterials-E636/).
>
> In these examples across various scenes, we compare the raw sampled frames with the corresponding image patches of our Affine Resampled Tokens. As shown in the visualizations, while the target objects undergo drastic displacement and perspective distortion in the raw images due to camera movement, the resampled token patches consistently maintain a stable position and pose.
>
> ### ebXC-Q1
> > Isolate the VGGT features.
>
> The contribution of our explicit affine mechanisms is indeed isolated. Please refer to our detailed **response to W1** for further clarification.
>
> ### ebXC-Q2
> > GPU memory and Inference latency.
>
> We appreciate your practical perspective on computational efficiency. We evaluate the inference latency and GPU memory allocation (in FP16 precision) on a single A100 (80GB) GPU. The results are summarized below.
>
> | Models | GPU Memory (GB) | Latency (ms/token) |
> | :--- | :--- | :--- |
> | Qwen2.5-VL-7B | 15.2 | 35.0 |
> | VG-LLM-8B | 17.1 | 36.5 |
> | AffIn-Space-8B | 17.3 | 36.8 |
>
> Thus, AffIn-Space introduces **negligible overhead** compared to existing 3D-aware MLLMs like VG-LLM.
>
> ### ebXC-Q3
> > Choice of the rank r=8 of the pose-correlated subspace.
>
> The choice of r=8 is theoretically driven by the degrees of freedom (DoF) of the affine transformations. As detailed in **§3.2**, the decomposed parameters (scale, two rotations, tilt, translation) span a manifold of roughly 6 DoF. A rank of r=8 provides just enough capacity to capture this pose-correlated noise. A smaller rank cannot cover the full DoF, while a significantly larger rank risks stripping away critical visual semantics.
>
> ### ebXC-Q4
> > Generalization to outdoor scenes.
>
> Thanks for your attention to the generalization of our work! Please refer to our **response to W2**.
>
> Overall, thank you for the careful review. We will incorporate a dedicated discussion in the camera-ready version.

---

> > ### Author Rebuttal · Reviewer_ebXC · 2026-04-01
> >
> > I thank the authors for their thorough rebuttal. My concerns have been adequately addressed.

---

> > > ### Author Response · Authors · 2026-04-04
> > >
> > > We sincerely thank you for your response and for acknowledging that your concerns are fully resolved. We are glad that our explanation addressed your points effectively.
> > >
> > > As you suggested, we will include all the clarifications and discussions from this rebuttal in the final camera-ready version of our paper. Thank you again for your time and for helping us improve our work.

---

### Decision · Program_Chairs · 2026-04-30

**Decision:**

Accept (regular)

**Comment:**

This paper proposes a new method for spatial understanding in MLLMs that reduces “geometric fragility”. The method is novel and intuitive, and the paper demonstrates strong, consistent improvements over solid baselines. Reviewers find the work compelling and all recommend acceptance. I agree with them.